# Exploring the Potentiality of a Plant Platform for Monoclonal Antibody Production in Veterinary Medicine

**DOI:** 10.3390/vaccines12060620

**Published:** 2024-06-04

**Authors:** Bertrand Morel, Claude Favrot, Lucie Mirande, Clemens Grünwald-Gruber, Virginie Stordeur, Louis Philippe Vezina, Loïc Faye, Véronique Gomord

**Affiliations:** 1ANGANY Innovation, 1 Voie de l’Innovation, 27100 Val de Reuil, France; bertrand.morel@angany.com (B.M.);; 2Dermatology Unit, Clinical for Small Animal Internal Medicine, Vetsuisse Faculty, University of Zurich, 8006 Zurich, Switzerland; 3BOKU Core Facility Mass Spectrometry, BOKU University, Muthgasse 18, A-1190 Vienna, Austria; 4ANGANY Inc., Suite 200, 873 St-Jean, Québec, QC G1R 1R2, Canada

**Keywords:** cytokines, pruritus atopic dermatitis, IL-31 inhibition, monoclonal antibody (mAb), plant-made antibody, plant-made pharmaceutical

## Abstract

Canine atopic dermatitis (CAD) is an allergic, inflammatory, and pruritic skin disease associated with the production of IgE antibodies against environmental allergens and mainly house dust mite allergens. This complex dermatological pathology involves Interleukin 31 (IL-31) as a central itch mediator. One of the most effective CAD treatments is a caninized monoclonal antibody (mAb) called Lokivetmab. It is produced in CHO cells and targets specifically canine IL-31 (cIL-31) and blocks its cellular messaging. This treatment has undoubtedly contributed to a breakthrough in dermatitis-related pruritus. However, its production in mammalian cells requires time-consuming procedures, high production costs, and investment. Plants are considered an emerging protein production platform for recombinant biopharmaceuticals due to their cost-effectiveness and rapidity for production. Here, we use transient expression in *Nicotiana benthamiana* plants to produce recombinant canine Interleukin 31 (cIL-31) and an anti-IL-31 monoclonal antibody (M1). First, we describe the production and characterization of M1 and then its activity on an IL-31-induced pruritic model in dogs compared to its commercial homolog. Dogs treated with the plant-made M1 mAb have shown similar improvements to Lokivetmab-treated ones after different challenges using canine IL-31. Furthermore, M1 injections were not associated with any side effects. These results demonstrate the safety and efficacy of this plant-made Lokivetmab biosimilar to control dogs’ pruritus in a well-established model. Finally, this study shows that the plant-production platform can be utilized to produce rapidly functional mAbs and bring hope to the immunotherapy field of veterinary medicine.

## 1. Introduction

Active and passive immunotherapies constitute a cornerstone of human therapies of cancers and chronic diseases, but they have not been fully developed in veterinary medicine yet. Veterinarians, scientists, and dog owners hope that the strong interest in immunotherapy will benefit canine health. Allergen-specific immunotherapy (desensitization) has been used in veterinary medicine for many decades to treat environmental allergen-induced atopic dermatitis [1,2,3,4,5]. The development of passive immunotherapy using monoclonal antibodies is, on the contrary, just starting in veterinary medicine, but some major progress has been made using such an approach for the treatment of pruritus, chronic pain, and some cancers [6,7,8,9,10].

Atopic dermatitis (AD) is a chronic, multifactorial condition resulting in inflamed and pruritic skin. This skin disease is the most common in dogs, with a prevalence of 3 to 15% in the canine population [11]. The development of pruritus in canine atopic dermatitis (CAD) is induced by several endogenous mediators, one of them being Interleukin 31 (IL-31) [12,13]. Lokivetmab is a caninized monoclonal antibody (mAb) specifically targeting IL-31 [14,15,16,17]. Blockade of IL-31 by Lokivetmab prevents IL-31 from binding to its receptor and therefore inhibits IL-31-mediated cellular messaging, providing relief from dermatitis-related pruritus for at least 4 weeks after a single subcutaneous injection (1 mg/kg) in atopic dogs [15]. Recently, an efficacy enhancement has been shown by combining Lokivetmab with topical therapies containing plant extracts and designed to repair the skin barrier [18]. Despite the remarkable results of this therapeutic approach, there are very few examples of passive immunotherapy in veterinary medicine. The scarcity of veterinary Mab mainly comes from the general production cost of mAbs in mammalian cells and therefore highlights the need for new platforms to produce them at a lower cost. For instance, the current treatment of a 10 kg dog with a monthly injection of Lokivetmab costs at least USD 1200 per year, as detailed by the *Barron’s* magazine, the world’s premier investing publication providing financial news.

It has been shown that various types of biologics could be produced by plants, including antibodies, vaccines, therapeutic proteins, and cytokines [19,20,21]. Plant expression platforms are safe and cost-effective. In addition, the production is easily scalable. These advantages make possible fast and global-scale deployment of biologics and other valuable recombinant proteins [19,22,23]. For all of these attributes, plants are considered an efficient alternative to the traditional expression systems, such as bacterial, yeast, insect, and mammalian cells. With more than 50 different antibodies expressed in various plant expression systems since the first report of antibody production in *N. tabacum* plants [24], the production of mAbs in plants using transient expression is well-documented [23,25]. Taking advantage of the rapid transient expression in *Nicotiana benthamiana* (*N. benthamiana*), we have produced and characterized recombinant canine IL-31 (cIL-31) and a plant-made Lokivetmab biosimilar (M1). We have shown that the interleukin produced in plants (rcIL-31 (Plant)) has identical biophysical properties to cIL-31 produced in HEK (rcIL-31 (HEK)), and its activity is fully confirmed with the induction of dog pruritus. Then, the safety and efficacy of the monoclonal antibody (M1) to control dog pruritus is demonstrated in this proof-of-concept study, as well as the clearly established plant expression platforms as a promising and powerful alternative for the production of therapeutic treatments in veterinary medicine.

## 2. Materials and Methods

### 2.1. Animals

All in vivo experiments were performed at the Bio2M facility (Mezilles, France) using (2–3-year-old) beagle dogs of both genders. All animal procedures were conducted in accordance with the standard operating procedures and the animal health regulations according to French laws. This study was coded CE22-05 by the BIO 2M ethics committee during its favorable evaluation authorized by the French Ministry of Research (26 December 2022) with the APAFIS code #39507-202211211340769 v5.

The general appearance and behavior of the animals were observed at least once a day throughout the animal phase from inclusion and recorded on a ‘Daily monitoring’ sheet.

Health indicators, such as weight change, skin, general condition of the animal, appetite, temperature, stools, blood count, and blood biochemical parameters, were monitored. A clinical examination was carried out every month (D-23, D1, D30, D61).

### 2.2. Molecular Design and cDNA Assemblies

For the expression of soluble recombinant canine IL-31 (rcIL-31 (Plant)), the nucleotide sequence encoding the mature IL-31 (accession number C7G0W1) was fused at its N-terminus to the tobacco chitinase signal sequence (accession number: QEQ12695) and at its C-terminal end to a hexa-histidine TAG (Appendix A).

For expression of the M1 monoclonal antibody, two constructs were used (Appendix A). The first one encoding for heavy chain (HC) contains HC variable region as described previously (SEQ ID NO 31; CAN-34D03-VH-568-1) and HC constant region (accession number: AAL35302.1). The second one encoding for light chain (LC) contains the previously described LC variable region (SEQ ID NO: 25; CAN 34D03-VL998-1) and LC constant domain kappa (XP_035562095.1). Each one was fused at the C-terminus to the tobacco chitinase signal sequence (accession number: QEQ12695).

The sequences of all cDNAs described above were optimized for expression in *Nicotiana benthamiana*.

### 2.3. Preparation of Plasmids

*XbaI*/*KpnI* and *SalI*/*SacI* restriction sites were, respectively, cloned at the 5′ and 3′ ends of each cDNA assembly described above. Then, the cDNAs were cloned into the binary expression vector pAG01 using these restriction sites [26]. Subsequently, *Agrobacterium tumefaciens* strain LBA4404 was transformed using the resulting vectors.

### 2.4. Plant Cultivation and Transient Expression

Wild-type or *FucT*/*XylT* mutant (ΔXF) [27] seeds from *N. benthamiana* were sown in coco fiber plugs. The seedlings were grown in a hydroponic system under continuous LED lighting. After 14 days, they were transferred to larger hydroponic tanks containing a nutrient medium under LED lighting at 26 °C and a 16 h/8 h day–night regimen for a further 14 days of development. At the end of this period, the aerial part of the tobacco plants was immersed in a suspension of agrobacteria carrying the inoculum. The inoculum was then infiltrated into leaves through two cycles of vacuum (−0.8 Bar)/release (1 min each cycle). Following vacuum infiltration, plants were transferred to hydroponic tanks for 4 days. The FucT/XylT mutant seeds were provided as a generous gift from Prof. Dr. Stefan Schillberg (Fraunhofer IME, Aachen, Germany).

### 2.5. Extraction and Purification of Recombinant Canine IL-31 in Plants (rcIL-31 (Plant))

Plant crude extract was prepared through passage of *N. benthamiana* leaves through a stainless steel juicer (Angelia 5500). After filtration on a 30 µm nylon filter, extract pH was adjusted to 8.2 using a 2M Tris solution. Magnetic beads, based on IMAC Sepharose medium with magnetite incorporated and nickel immobilized (His Mag Sepharose Excel, Cytiva, Malborough, MA, USA) at a volume of 5 mL of sedimented magnetic gel, were directly introduced into the unclarified crude extract. After 2 h of incubation with constant slow rotation on a rotator, the magnetic beads were recovered and washed with 1 L of 50 mM of Tris pH 8.2 and 250 mL of 50 mM of Tris with 10 mM of imidazole pH 8.2. Then, rcIL-31 (Plant) was eluted from the magnetic beads with 50 mL of 50 mM of Tris with 200 mM of imidazole pH 8.2. Fractions containing the recombinant protein were pooled and submitted to an ion exchange chromatography on a Hi Trap Q FF column (Cytiva, Malborough, MA, USA) equilibrated in 50 mM of Tris pH 8.2 at a flow rate of 0.5 mL/min. The flow through was then concentrated on an Amicon Ultra 3 kDa centrifugal unit (Merck, Darmstadt, Germany). Size-exclusion chromatography (SEC) was performed on HiPrep™ 16/60—Superdex™ 200 HR (Cytiva, Malborough, MA, USA) equilibrated in 50 mM of Na phosphate and 150 mM of NaCl pH 7.5 (PBS). The SEC fractions containing rcIL-31 (Plant) were subsequently pooled, sterilized through filtration on a 0.22 µm filter, and kept at −20 °C until further use.

### 2.6. Extraction and Purification of a Recombinant Monoclonal Antibody Specific for cIL-31 (M1)

*N. benthamiana* leaves were harvested 4 days post-infiltration and extracted using a juicer (Angelia 5500). The pH of the crude extract was adjusted to 7.5 through the addition of 2M of Tris and centrifuged at 12,000× *g* (10 °C) for 40 min. Subsequently, the clarified supernatant was submitted to filtration using a 750 kDa hollow fiber module (Repligen, Whaltham, MA, USA). The resulting filtrate was loaded at a flow rate of 15 mL/min onto a Hi Screen Fibro PrismA (Cytiva, Malborough, MA, USA) equilibrated in 50 mM of Tris with 125 mM of NaCl pH 7.5. After washing in the binding buffer, the antibody was eluted with 0.1 M of Glycine-HCl at pH 3.2, and the eluate was immediately neutralized with 1M of Tris. Purified plant-produced M1 antibody was concentrated using an Amicon Ultra 30 kDa centrifugal unit (Merck, Darmstadt, Germany) and buffer-exchanged on PD10 desalting columns (Cytiva, Malborough, MA, USA) with formulation buffer at pH 6.2, containing 5 mM of L-His, 6.2 mM of His-HCl, 2.6 mM of L-met, 0.06 mM of EDTA, 105 mM of Trehalose, and 0.2% Tween 80. Finally, the purified antibody was 0.22 μm filter sterilized before storage at 4 °C.

### 2.7. SDS-PAGE and Western Immunoblot Analyses

Protein samples were analyzed using SDS-PAGE gels after heating at 90 °C for 5 min in denaturation buffer containing Tris 62.5 mM, pH 6.8, 10% glycerol, 1% SDS, and 2% βmercaptoethanol. For Western blot experiments, gels were transferred onto a nitrocellulose membrane (InVitrogen iBlot2 gel transfer Transfer Device (Thermofisher, Waltham, MA, USA, IB21001)) using an InVitrogen iBlot2NC regular stack (Thermofisher, IB23001). The primary antibodies used for cIL-31 immunodetection were either a polyclonal rabbit-antisera provided by AgroBio (La Ferté Saint-Aubin, France) and directed against cIL-31 at a 1:5000 dilution followed by a secondary goat anti-rabbit IgG antibody coupled to horseradish peroxidase (Bio-Rad, Hercules, CA, USA) at a 1:30,000 dilution or the monoclonal antibodies Lokivetmab or M1 at a concentration of 0.2 μg/mL and followed by a secondary anti-dog IgG antibody at a 1:15,000 dilution (Antibodies, St. Peters, MO, USA, ABIN101093). The plant-made monoclonal antibody (M1) was immunodetected using anti-dog IgG produced in rabbit (Sigma-Aldrich, St. Louis, MO, USA) at a 1:5000 dilution followed by a secondary goat anti-rabbit IgG antibody coupled to horseradish peroxidase (Bio-Rad, Hercules, CA, USA). Western blots were visualized with chemiluminescent substrate (SuperSignal West Pico Plus, Thermofisher) and an Invitrogen iBright 1500 imaging system.

### 2.8. Dynamic Light Scattering

The apparent hydrodynamic diameters of rcIL-31 and M1 were measured using dynamic light scattering (DLS) on a Malvern Zetasizer Pro (Malvern Panalytical, Malvern, UK). Malvern ZS Xplorer software v3.0 was used in data collection and processing. Sets of DLS data were measured at different temperatures to control thermal stability and performed 3 times with 10 runs of 30 s each.

### 2.9. Size Exclusion Chromatography

The Akta Pure (Cytiva, Malborough, MA, USA) purification system was used to assess protein purity and aggregation. In brief, antibody samples (1 mg/mL, 1 mL) were injected into the Superdex 200 Hi Load 16/600 column (Cytiva, Malborough, MA, USA) at a flow rate of 1 mL/min. The absorbance of the elution peak was monitored at 280 nm.

### 2.10. N-Linked Glycan Analysis

Mass spectrometry was used to compare the N-glycosylation profiles of Lokivetmab produced in CHO cells and M1 glycovariants produced either in wild-type or in a FX-Ko line of *N. benthamiana* [27].

The proteins were initially subjected to in-solution digestion, followed by S-alkylation using iodoacetamide, and then further digested with Trypsin from Promega. Subsequently, deglycosylation was carried out using PNGase A to assess the proportion of non-glycosylated peptides, and the samples were analyzed via LC-MS. The digested samples were injected onto a nanoEase C18 column (nanoEase M/Z HSS T3 Column, 100 Å, 1.8 µm, 300 µm × 150 mm, Waters) with 0.1% formic acid as the aqueous solvent. A gradient ranging from 1% B (B: 80% Acetonitrile, 0.1% FA) to 40% B over 50 min was employed, followed by a 10 min gradient from 40% B to 95% B to aid in the elution of larger peptides, at a flow rate of 6 µL/min. Detection was performed using an Orbitrap MS (Exploris 480, Thermo, Waltham, MA, USA) equipped with the standard H-ESI source in positive ion, DDA mode (switching to MS/MS mode for eluting peaks). MS scans were recorded in the range of 350–3200 Da, and the top 10 peaks were selected for fragmentation. Instrument calibration was carried out using Pierce FlexMix Calibration Solution (Thermo Scientific, Waltham, MA, USA). The putative glycopeptides were identified as sets of peaks comprising the peptide portion and the associated N-glycan, with variations in the number of HexNAc units, hexose, deoxyhexose, and pentose residues. The theoretical masses of these glycopeptides were calculated using a spreadsheet based on the monoisotopic masses for amino acids and monosaccharides. Manual glycopeptide searches were conducted using Freestyle 1.8 (Thermo). For quantification of the different glycoforms, the peak areas of Extracted Ion Chromatograms (EICs) of the first four isotopic peaks were summed, employing the quantification software Skyline (version 21.1.0.278, MacCoss Lab Software, University of Washington, Seattle, WA, USA).

### 2.11. Antibodies Binding to cIL-31 According to Biolayer Interferometry

The affinity between mAbs and canine Interleukin 31 was assessed using the Sartorius Octet R8 Bio-layer Interferometry (BLI) equipment (Sartorius Lab instruments, GmbH & Co. KG, Goettingen, Germany). Experiments were performed at 25 °C in PBS at pH 7.4, supplemented with 0.1% BSA and 0.02% Tween. Lokivetmab or M1, at a concentration of 5 nM, was immobilized onto ProA biosensors (Sartorius, Tokyo, Japan, 18-5010) until a BLI signal response of 1 nm was achieved. To measure association rates, the loaded biosensors were transferred to wells containing rcIL-31 (HEK) at different concentrations ranging from 300 to 80 nM. Dissociation rates were determined by dipping the biosensors into buffer-containing wells. The association duration was set at 360 s, followed by a dissociation period of 600 s. Data were recorded and analyzed using Octet BLI Discovery v13.0 and Origin Pro 8.5 (Originlab, Northampton, MA, USA).

### 2.12. Interaction between Neonatal Fc Receptor and Antibodies Using Bioluminescent Immunoassay

The experiments were performed using the Promega kit (Ref CS3019A02) and following the described protocol.

This assay was divided into three steps. First, 25 μL of IgG-LgBiT solution was mixed with 25 μL of the monoclonal antibody (Lokivetmab or M1) in a white, 96-well, non-binding plate (Corning, New York, NY, USA). Subsequently, 50 μL of cFcRn-SmBiT was added to each well containing IgG and IgG-LgBiT, and plate was incubated for 30 min at room temperature. Then, 25 μL of Lumit detection solution and signal was allowed to stabilize for 3–5 min. Subsequently, the bioluminescence signal (relative light units [RLU]) was measured in a Victor Nivo plate reader (Perkin Elmer, Turku, Finland). All of the dilutions were made in PBS containing 10% of adjusting buffer to decrease the final pH to a value of 6. All of the experiments were run in duplicate or triplicate. Normalized RLU data were generated by assigning 100% to the maximum bioluminescent signal obtained in absence of the IgG and then calculating percentage drop in the signal in the presence of an IgG. To obtain the inhibition curves, normalized RLU as a function of IgG concentration (μg/mL) log value was plotted. Experimental data were fitted to a four-parameter logistic equation using Origin Pro 8.5 (OriginLab).

### 2.13. Challenges and Injection Regimen

Laboratory beagle dogs were acclimated during six weeks and subjected to physical examination, body weight measurement, blood collection (hematology and clinical chemistry), and daily clinical observations. At the end of the acclimation phase, 12 dogs were selected for inclusion into the testing phase. Altogether, 6 females and 6 males in perfect health and aged 2 to 3 years were enrolled. Prior to mAb injection, each dog was challenged three times within an interval of 1 week with a cIL-31 dose of 4 μg/kg administered intravenously. Before mAb injection, comparison of the level of pruritus (in seconds per hour over two hours) between baseline pruritus (D-22 and D-23) and the cIL-31 challenges (D-20, D-14, and D-7) was performed for each dog. Dogs were then randomized into two groups of 6 dogs. One group received M1, and the other one had Lokivetmab administered subcutaneously at a dose of 1.5 mg/kg.

Pruritus was induced upon injection of cIL-31 (HEK) (4 μg/kg) after the mAb injection and also one month (D30) and two months after (D61). Level of pruritus (in seconds per hour over two hours) recorded at D1, D30, and D61 were compared to pruritus values recorded after cIl-31 (HEK) challenges at D-20, D-14, and D-7 preceding mAb injection.

### 2.14. Quantification of cIL-31-Specific IgG

Quantification of cIL-31-specific IgG was achieved one month after mAb injection through indirect ELISA using reference dog IgG as the standard curve (BioRad, Hercules, CA, USA). ELISA plates were coated with 50 ng of the recombinant cIL-31 or a three-fold serial dilution of dog IgG spanning a range from 1500 ng/mL to 11.7 ng/mL in a carbonate buffer pH 9.6 overnight at 4 °C (100 µL/well). Between each step prior to the addition of the substrate, plates were washed thoroughly with a solution of TBS-Tween-20 0.1% (TTBS). The wells were blocked to prevent non-specific binding with 200 µL of blocking buffer (5% skimmed milk powder in TTBS) for 2 h at a controlled temperature of 37 °C. Subsequently, diluted serum samples in 2.5% milk in TTBS were added to the wells (100 µL/well), and dilution buffer alone was added to the standard curve. Plates were incubated for 2 h at 37 °C. Anti-dog IgG coupled to horseradish peroxidase (HRP) (Antibodies, ABIN101093) was diluted 1:30,000 in the dilution buffer, added to the wells (100 µL/well), and incubated for 1 h at 37 °C. O-phenylenediamine dihydrochloride (Sigma-Aldrich, St. Louis, MO, USA) substrate (200 µL/well) was used for signal detection, and the reaction was quenched with H_2_SO_4_ 2M (100 µL/well) after 30 min of incubation at 16 °C in the dark. Absorbances at 485 nm were recorded on a Victor Nivo microplate reader (Perkin Elmer, Turku, Finland).

## 3. Results

### 3.1. Production, Purification, and Biological Activity of Plant-Made rcIL-31 (Plant)

Following molecular cloning and agrobacterium transformation, agroinfiltration was used to deliver the pAG01 vector carrying cIL-31 cDNA into the plant cells for transient expression. However, after centrifugation of the crude extract, and probably due to interactions with plant cell debris, most rcIL-31 (Plant) did not stay in the supernatant, and clarification of the plant extract could not be performed before purification. To solve this issue, we used magnetic purification by adapting the protocol described recently in Faye et al. [28]. Ni magnetic bead slurry was added directly to the plant crude extract. After two hours of incubation, the magnetic beads were washed in 50 mM of Tris pH 8.2 and in 50 mM of Tris with 10 mM of imidazole pH 8.2 prior to the elution of rcIL-31 (Plant) in 50 mM of Tris with 200 mM of imidazole pH 8.2. Further purification was performed using ion exchange chromatography, and a polishing step using size exclusion chromatography was also carried out. Integrity and purity of rcIL-31 (Plant) were confirmed through Western blotting and silver staining gel analyses (Figure 1A).

DLS experiments were performed to determine the hydrodynamic radius and the oligomeric state of cIL-31 at pH 7.5. As shown in Figure 1B, the hydrodynamic radius (R_h_) appears to be between 2.7 and 2.9 nm. The size distribution profiles are similar whether cIL-31 was produced in plants or in HEK cells, and no large aggregates are detected. Therefore, there is no significant difference in size between plant and HEK cIL-31, and the polydispersity similarity indicates a comparable characteristic in terms of structure compactness. However, the R_h_ value is larger than expected for a protein with such molecular weight and is rather consistent with a dimeric form. This dimeric nature was also confirmed with the elution volume obtained from the analysis on a calibrated Superdex 200 column and is in accordance with the oligomeric state previously reported for some interleukins [29].

Then, the biological activity of the plant-produced cIL-31 was tested on animals. This was performed by measuring its capacity to induce dogs’ pruritus. Pruritus duration was monitored for two hours after administration of a 4 μg/kg dose on 12 dogs (Figure 1C). To compare the activities between plant-made cIL-31 and cIL-31 produced in HEK cells, both products were injected into the dogs at different time intervals. From this experiment, we conclude that cIL-31 produced in plants is fully active and induces pruritus in dogs in a similar way to cIL-31 produced in HEK cells.

### 3.2. Production, Purification, and Characterization of the Plant-Made Anti-cIL-31 Monoclonal Antibody

Here, ΔXF *N. benthamiana* plants were co-infiltrated with *Agrobacterium* harboring the heavy chain and light chain vectors of the antibody mixed at a ratio of 1:1. The infiltrated leaves were collected when the highest expression rate was reached, i.e., 4 days after vacuum infiltration (dpi). After centrifugation and clarification of the green juice obtained from plant extraction, M1 was affinity purified using Protein A affinity chromatography. The absence of signal in the flow through (FT) on the Western blot analysis (Figure 2A) indicates that the recovery is good, and the antibody was eluted properly with an estimated recovery yield close to 100%.

The purified antibody was buffer-exchanged and then analyzed for integrity and purity using SDS-PAGE. Under non-reducing conditions, a band corresponding to the fully assembled antibody ≈ 150 kDa was observed (Figure 2B). The heavy chain (C_H_) migrating at ≈50 kDa and the light chain (C_L_) at ≈25 kDa were observed when SDS-PAGE was performed under reducing conditions (Figure 2B). In Figure 2B, electrophoretic analysis of different batches of M1 is presented, and this reveals a very good repeatability and reproducibility of the purification process of this monoclonal antibody produced in plants. Although minor amounts of a few additional bands corresponding to antibody fragments are present in M1 (Figure 2C), M1 is obtained with high purity following a fast and straightforward purification method.

Additionally, purity and aggregation states were further characterized using size exclusion chromatography and DLS experiments. The results demonstrated that plant-made antibody M1 assembled as full IgG molecule (major peak), as shown in Figure 2D, and the SEC elution profile is very similar to the one obtained for Lokivetmab. Subsequently, DLS experiments have shown that M1 and Lokivetmab have similar hydrodynamic radii (R_h_) and that no aggregates are detected (Figure 2E). Although a small shift in R_h_ upon an increase in temperature occurs, both antibodies appear to have similar thermal stabilities (Figure 2E).

### 3.3. N-Linked Glycosylation Patterns of M1 and Lokivetmab

The mass spectra of the glycosite (EEQFNGTYR) of M1 and Lokivetmab are shown in Appendix A. The peak heights in the MS spectra roughly reflect the molar ratios of the glycoforms, and the quantitation of these different glycoforms is shown (Appendix A). Please note that the method used can only elucidate the composition of the glycans, but not the glycan structures. The indicated structures only represent one plausible composition. The ProGlycan nomenclature was used (homepage.boku.ac.at/jstadlmann/nomenclature.html). The mAb produced in CHO cells showed a single predominant N-glycoform, GnGnF (79%), and less than 1% of high mannose N-glycans. In contrast, oligomannosidic N-glycans are very abundant (44%) in M1 produced either in the WT or FX-KO line of *N. benthamiana*. Major glycoforms identified in M1 WT were complex type glycans (GnGn(F)(XF), (37.2%). Consistent with previous results [28], no α(1,3) fucosylated or β(1,2) xylosylated N-glycans were detected in M1 produced in the FX-KO line of *N. benthamiana*, thus confirming a complete knock-out of β(1,2) xylosyltransferases and α(1,3) fucosyltransferases genes in this mutant. M1 and Lokivetmab also contained a small percentage of non-glycosylated peptide.

### 3.4. In Vitro Functional Characterization of Plant-Made Antibody

One of the critical quality attributes to be investigated when developing an alternate antibody format is its interaction with the target antigen.

Binding capacity of the plant-made monoclonal antibody (M1) to rcIL-31 (HEK) was first assayed on Western blots and compared to the one of Lokivetmab (Figure 3A). Both mAbs have similar reactivity, as shown by almost identical band densitometry.

Next, we checked whether the antibody produced in the plant had the same function as Lokivetmab and thus compared the affinities with canine IL-31 obtained for these two antibodies through biolayer interferometry (BLI). The dissociation constant (K_D_) of either M1 or Lokivetmab with the target antigen (rcIL-31 (HEK)) was determined using the ratio of the dissociation (k_d_) and association (k_a_) rate constants. The binding properties of the antibodies were studied with four different antigen concentrations. Then, the kinetic rate constants were derived using a global fit analysis. The results show that binding to canine IL-31 is concentration-dependent (Figure 3B,C). Although the dissociation constant is slightly smaller for Lokivetmab, the values are not significantly different. Indeed, the plant-made antibody was found to exhibit a comparable binding affinity (K_D_ = 28.5 nM) to its homolog Lokivetmab (K_D_ = 27.44 nM) against rcIL-31 (HEK). The obtained K_D_ values are within the expected values for an antigen–antibody interaction and reveal a strong interaction between cIL-31 and the two monoclonal antibodies, thereby validating the in vitro activity of our plant-made antibody. Although the dissociation constants are identical, it is interesting to note that the association rate constants, derived from our BLI experiments, are different between the two antibodies. This could be correlated with the difference in the glycosylation pattern and the steric hindrance effect of slightly larger glycans.

### 3.5. Binding to FcRn

The neonatal Fc receptor (FcRn) plays an important role in immune characteristics through, essentially, its capacity for IgG lifetime modulation. Therefore, it determines mAb efficacy [30,31] and relates to the availability of the therapeutic antibody in the body.

Here, we used a luminescence assay to compare the FcRn binding to the therapeutics antibodies (Figure 4). Lumit canine FcRn immunoassay does not require any washing procedure and is based on a competition assay to measure the interaction between canine FcRn and IgG. This experiment involves a canine IgG labeled with LgBit (IgG-LgBit) acting as the tracer. The target molecule consists of a C-terminus biotinylated cFcRn attached to Streptavidin (cFcRn-SmBiT). The maximum luminescence signal is obtained for the sample that does not contain IgG. For samples containing canine Fc domain, the samples will compete with IgG-LgBiT tracer to bind to the target, resulting in a concentration-dependent decrease in the luminescent signal.

It seems that the presence of xylose and fucose does not have a significant impact on the affinity of M1 with FcRn, as shown by the similar IC_50_ values obtained for M1 WT and M1 ΔXF.

M1 variants have slightly lower values of K_d_ (18–22 μg/mL) than Lokivetmab (K_d_ = 45 μg/mL). This small difference could possibly come from the differences in the N-glycosylation patterns of plant-made M1 and CHO-made Lokivetmab, as shown in the Appendix A. Nevertheless, the values remain in the same order of magnitude, and this could indicate that the pharmacokinetics and IgG half-life of the plant-made mAbs could be very similar, or even slightly better, than the ones of its analog produced in CHO cells.

We have successfully produced functional anti-IL-31 mAbs in *N. benthamiana* through transient expression. The plant-produced mAb showed specific binding to the canine IL-31 and FcRn. However, before a plant-produced antibody can be used clinically, further studies in animal models should be carried out to test the safety, pharmacokinetics, and efficacy.

### 3.6. M1 Efficacy in Controlling Dog Pruritus

In this study, the capacity of M1 to control dog pruritus was monitored and compared to Lokivetmab (Figure 5). Before immunotherapy, the level of pruritus (in seconds per hour over two hours) between baseline pruritus at D-23 and D-22 and cIL-31 challenges (D-20, D-14, and D-7) is compared for each dog. Only dogs with pruritus values after IL-31 challenge (at D-20, D-14, and D-7) equivalent to twice the average of the basal values (evaluated at D-23 and D-22) and in good health were included in the study.

The selected dogs were separated into two groups. Six dogs were treated on Day 0 with the plant-made mAb (M1) produced in FX-Ko lines of *N. benthamiana*, while six others received Lokivetmab (Figure 5).

The same dose of each mAb (1.5 mg/kg) was administered to the 12 dogs. They were then challenged with rcIL-31 (HEK) at a dose of 4 μg/kg on Day 1, Day 30, and Day 60. At Day 1, pruritus control was more efficient for dogs treated with the plant-made antibody than for the ones treated with the homologous antibody produced in CHO cells (*p*: 0.02 vs. *p*: 0.07, Wilcoxon match-pairs rank test) when compared to the induced pruritus recorded before mAb injection (Figure 6A,B). It is also very important to note that injection of both mAbs was not associated with any side effects.

Furthermore, the in vivo stabilities of M1 and Lokivetmab were compared by analyzing mAb concentrations in sera collected 30 days after injection (D30). Mab quantification with ELISA reveals that the specific IgG concentrations are not statistically different even 30 days after the injections (Figure 6C).

These results indicate that in vivo stability of the plant-made antibody is the same as the one of the mAb produced in CHO cells, and they corroborate perfectly with the FcRn interaction results that have shown very close pharmacokinetics of the two drugs.

## 4. Discussion

Over the last decade, monoclonal antibodies (mAbs) and their high specificities have raised an important interest in the pharmaceutical industry, and these blockbusters are now a hot topic for the companies. Currently, most of the approved or commercialized mAbs are IgG antibodies that target cancer and auto-immune disorders, although there is a growing interest in monoclonal antibodies to tackle infectious diseases in human and veterinary therapies.

Generally, the therapeutic antibodies are mostly produced in mammalian cell cultures [32]. Their optimized manufacturing conditions and their regulation favor industrial use for recombinant protein production. However, this traditional expression system still has limiting factors, such as expensive raw materials, safety, and risk of contamination. This procedure requires time and tight in-process control to guarantee that the culture remains axenic, and that the growth curves at each step follow pre-established working specifications [33]. However, the high production cost of these drugs limits their prescriptions and restricts them to the most developed countries.

Many studies have illustrated that plants have the capacity to produce high-quality recombinant antibodies [19,34,35,36] and also biopharmaceuticals designed for new therapeutic approaches [37,38,39]. Therefore, plant-made mAbs have potential to be a viable alternative to their counterpart produced in mammalian cells [25]. The major advantage of the plant expression platform, such as the one supplied by ANGANY, compared to mammalian cell production is that the risk of contamination coming from human or animal pathogens is significantly reduced [40,41]. The different upstream stages in the mammalian cell platform require well-controlled/characterized activities. Nowadays, contract manufacturing organizations (CMOs) provide all of the upstream technologies, including storage, maintenance, and characterization of the master cell banks. However, this comes at a price and makes any company strongly dependent on the CMO for all of the life of the product, with no guarantee that the cost per lot charged will remain the same over the years.

All of these procedures are not necessary with the plant platform. Indeed, the inoculum is an agrobacteria strain easily characterized through full genome sequencing, and it is stored frozen. Therefore, the cost of production is drastically reduced compared with traditional biomanufacturing.

Another principal aspect of the plant technology for mAbs production is that it is relatively low-tech and requires little or no extraordinary laboratory facilities for production and characterization/storage of the Master cell bank and no costly cell culture facilities [25,42,43,44]. For these reasons, the time to build and implement a plant platform represents a major advantage compared to other facilities. This is a huge factor for animal pharmaceutical companies. For instance, if any of these companies want to install a plant-based factory, it would take 18 months to build, while other facilities for mammalian cell platforms would take between 5 and 7 years in addition to expansive installation costs. Finally, from a capital expenditures (CAPEX) point of view, the plant platform is appealing and could be an easy and affordable avenue for even large animal health pharmaceutical companies to access the mAb market or any other active pharmaceutical ingredients, with independence and total control of their costs.

The advantages of such a production platform are clearly illustrated by Kentucky Bioprocessing (KBP), which is a global leader in the expression, extraction, processing, and purification of biopharmaceutical proteins from plant systems. Working with a global network of collaborators, KBP uses its technology, know-how, and cGMP-compliant facilities to produce biopharmaceutical proteins for research and development, as well as preclinical and clinical applications. Of particular importance, Zmapp is a cocktail of three humanized and glycoengineered monoclonal antibodies against Ebola [35]. Remarkably, Zmapp was produced in 10 days using the KBP Good Manufacturing Platform (GMP) and was therefore rapidly administered to humans in various countries after FDA approval to face the pandemic situation.

Here, we have used transient expression in Nicotiana benthamiana plants to produce fully active canine cytokine IL-31 and a recombinant anti-IL-31 monoclonal antibody (M1). Dogs treated with the plant-made M1 mAb obtained identical or even slightly better control of IL-31-induced pruritus compared to Lokivetmab-treated ones after challenge with canine IL-31. M1 injections were not associated with any side effects. Serology performed after injection has shown that serum concentrations of both plant- and CHO-made mAbs were not statistically different, thus illustrating a very high stability of the plant-made M1-mAb.

Therefore, the present study demonstrates the safety, in vivo stability, and efficacy of the newly developed plant-based monoclonal antibody M1 to control dogs’ pruritus in a well-established model. Altogether, these data further confirm that plants could produce properly folded and fully active complex proteins. An historical argument against the use of plants as a production system for therapeutic proteins was their inability to perform mammalian-like N-glycosylation. Such differences in the N-glycosylation pathway, including the presence of β(1,2)-xylose and core α(1,3)-fucose, can affect the activity, potency, and immunogenicity of plant-derived proteins [45]. Extensive successful glycoengineering approaches resulted in the generation of glycoengineered *N. benthamiana* ∆XF plants where β(1,2)-xylosyltransferase and core α(1,3)-fucosyltransferase activities are knocked out [27]. This type of plant line is currently used to produce monoclonal antibodies that are completely devoid of core α(1,3)-fucose and β(1,2)-xylose residues and resemble human-type complex N-glycans [27,28,46,47,48]. Here, based on our glycosylation analysis (Appendix A), proteins purified from plant extract have a more homogeneous glycosylation profile compared to their CHO homologs. This observation should be the same for any kind of antibody produced with the same seeds. Here, M1 was produced in the FX-KO line of *N. benthamiana* described previously [27]. In contrast with M1 produced in wild-type *N. benthamiana*, glycans N-linked to the glyco-engineered mAb are completely devoid of core α(1,3)-fucose and β(1,2)-xylose residues, and the predominant glycoforms are GNGN (30%) and M5–M9 high mannose type N-glycans (44%). Compared with N-glycosylation of Lokivetmab, where the major glycoform is α(1,6) fucosylated, GNGNF (79%), the absence of fucose on M1 probably increases its in vivo stability, as shown in this work by the slightly higher affinity to canine FcRn and the similar pharmacokinetics demonstrated with the ELISA experiments. Indeed, fucose residues are known to compete for mAb binding to the neonatal receptor (FcRn).

## 5. Conclusions

In conclusion, *N. benthamiana* plant-cell-based platform feasibility was illustrated for the rapid expression and purification of the anti-Il-31 IgG designed for the treatment of canine atopic dermatitis. The plant-produced mAb has in vitro physicochemical and functional characteristics quite similar to mammalian-cell-produced Lokivetmab (Cyopoint^®^) and has revealed comparable in vivo efficacy.

Altogether, this proof-of-concept study proves the reproducibility, scalability, and cost-effectiveness of the plant expression system as an alternative platform for large-scale antibody manufacturing, which could be used for immunotherapy treatments not only for veterinary issues but also for global health strategies.

## Figures and Tables

**Figure 1 vaccines-12-00620-f001:**
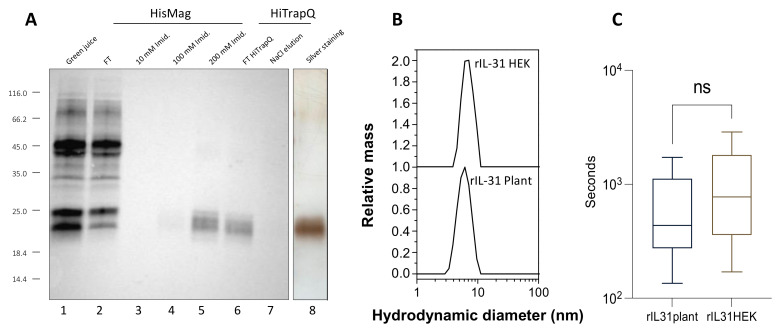
Expression, purification (**A**), oligomeric characterization (**B**), and pruritic activity (**C**) of recombinant cIL-31 produced in plant. (**A**) The protein extracts from leaves producing rcIL-31 were separated using SDS-PAGE and immunodetected with specific immunoserum for IL-31. The protein was then purified through different runs of chromatography, Ni affinity, ion exchange, and size exclusion chromatography. The purity of the final product is illustrated with the silver staining SDS PAGE gel. (**B**) Size distribution of the hydrodynamic radius measured through dynamic light scattering. (**C**) Analysis of the pruritogenic effect of soluble cIL-31 produced in recombinant form in *N. benthamiana* (rIL31 Plant) compared to the HEK-produced one (rIL31 HEK). After injection (IV) of rIL31, the dogs were monitored for scratching/itching, and duration was recorded for two hours.

**Figure 2 vaccines-12-00620-f002:**
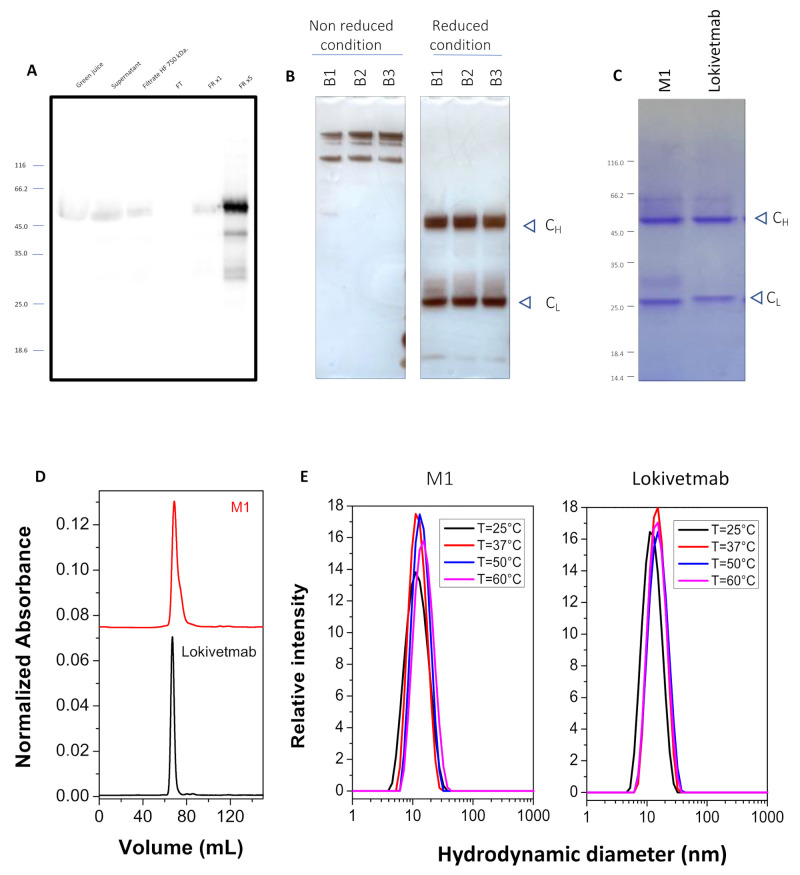
Characterization of the plant-made monoclonal antibody (M1). Analysis of the different purification stages through immunodetection using Western blot (**A**). (**B**,**C**) Purity of the purified antibody through silver or Coomassie staining. The high repeatability between three different batches (**B1**, **B2**, **B3**) is also shown (**B**). Oligomeric state and thermal stability comparisons between Lokivetmab and M1 studied through size exclusion chromatography (**D**) and dynamic light scattering (**E**).

**Figure 3 vaccines-12-00620-f003:**
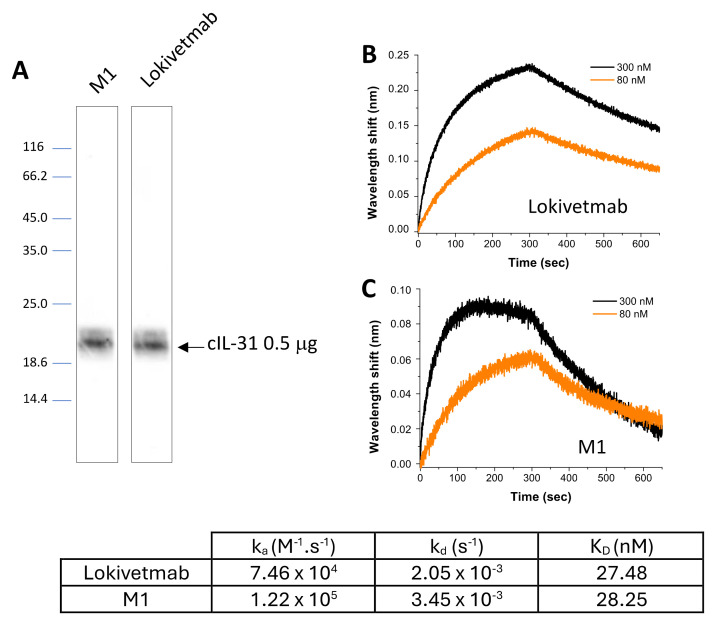
In vitro functional characterization of the antibodies. Comparison of the specificity of M1 and Lokivetmab with cIL-31 studied using Western blotting analyses (**A**). Concentration dependance of the association and dissociation rate constants of Lokivetmab (**B**) and M1 (**C**) with different concentrations of cIL-31 studied through biolayer interferometry. For the sake of clarity, association and dissociation results are shown for two concentrations.

**Figure 4 vaccines-12-00620-f004:**
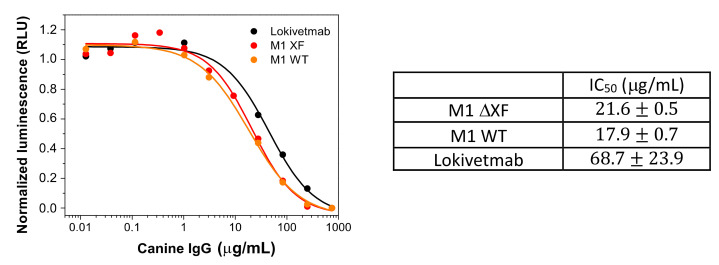
Interaction with canine neonatal Fc receptor through luminescence assay. Binding of M1 variants and Lokivetmab to the canine neonatal Fc receptor cFcRn as determined with a cFcRn-SmBiT luminescence assay. Normalized luminescence was calculated by assigning 100% to the maximum signal in the absence of an analyte and then calculating the percentage of maximum signal in the presence of an analyte. Mean IC_50_ values +/− standard deviation of two individual measurements is depicted.

**Figure 5 vaccines-12-00620-f005:**
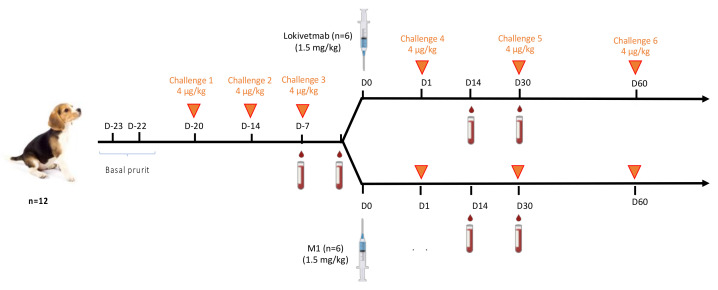
Schematic view of the immunotherapy and challenges regimen.

**Figure 6 vaccines-12-00620-f006:**
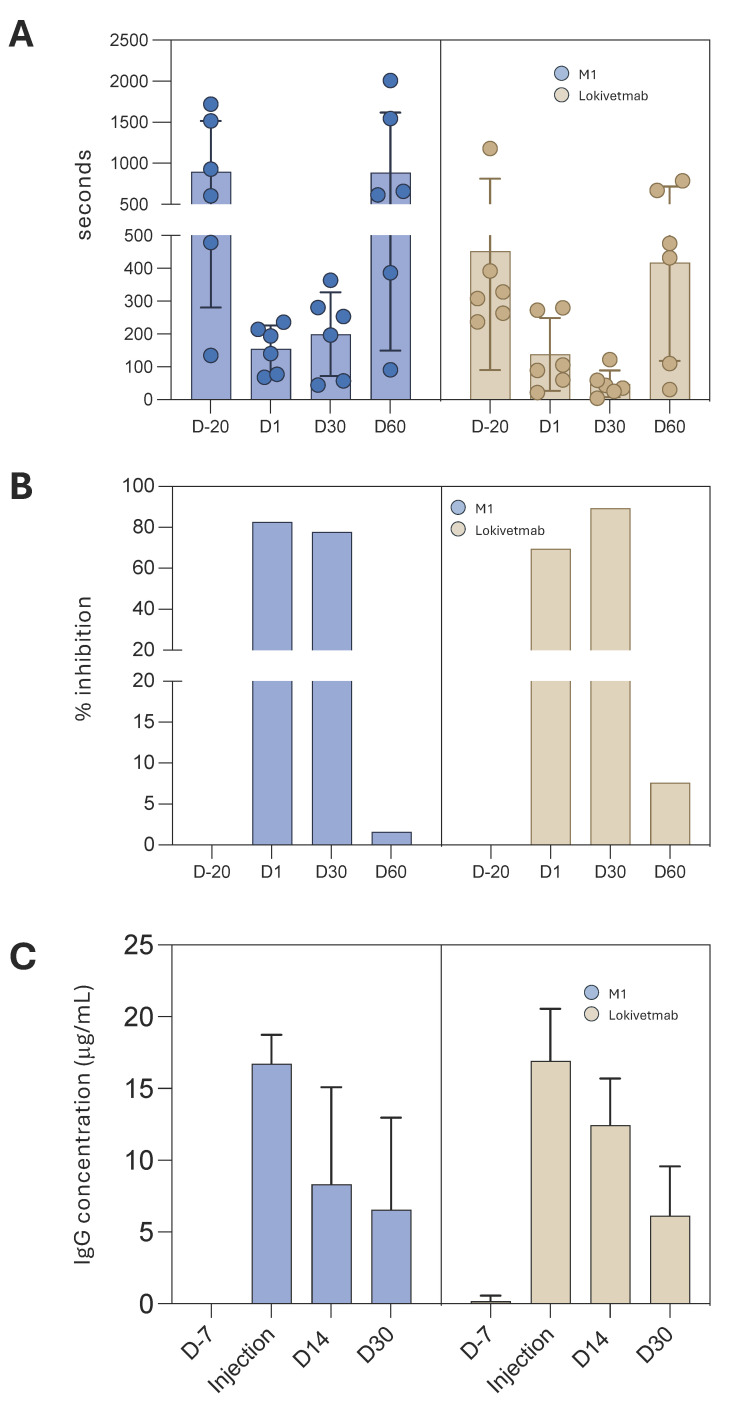
Comparison of the efficacy and stability of the antibodies. (**A**) Duration of the induced pruritus after the different challenges and percentage of inhibition (**B**). IgG concentration in the sera was monitored at different time points (injection, D14, and D30). hIgG concentration was determined using ELISA experiments (**C**).

## Data Availability

The data presented in this study are contained within this article.

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
