# Peer review of "Exploring the Potentiality of a Plant Platform for Monoclonal Antibody Production in Veterinary Medicine"

_vaccines, 2024, doi:10.3390/vaccines12060620_

Round 1
Reviewer 1 Report
Comments and Suggestions for Authors
Please see the attached pdf for detail comments

There needs to be a review of the english grammar in the manuscript - I have indicated some in the attached pdf
Author Response
We would like to thank this referee for all the valuable comments made on our manuscript.
- All the spelling and grammatical mistakes were corrected.
- Line 67-69 : we have rephrased the sentence to: “With more than 50 different antibodies expressed in various plant expression systems since the first report of antibody production in tabacum plants [23], the production of mAbs in plants using transient expression is well documented [22, 24].”
- We have added a Figure to represent the constructs in the Supplementary information(Figure S1a and Figure S1b).
- We have specified that the magnetic beads His Mag used for cIL-31 purification was specific to Histidines and added this sentence: “Magnetic beads, based on IMAC Sepharose medium with magnetite incorporated and nickel immobilized (His Mag Sepharose Excel, Cytiva, Malborough, MA, USA)”
- Figure legend number 1. We have corrected the mistake.
- Line 319-327: we have moved this section describing the state of plants glycosylation in the discussion part such as required.
- Line 333-335: we have changed the sentence about the recovery yield by: “The absence of signal in the flow through (FT) on the western blot analysis (Figure 2A) indicates that the recovery is good and antibodies were eluted properly with an estimated recovery yield close to 100%.”
- Line 390: We determine the Kd values with 4 different concentrations for the two antibodies. Two concentrations are represented in Figure 3 for clarity. We have added the following sentence in the figure legends: “For the sake of clarity, association and dissociation results are shown for two concentrations.”
- We agree that although the dissociation constant is identical, there is a difference in the association rate of the two antibodies. This difference could come from a difference in the glycans nature affecting the steric hindrance.
We have added this sentence: “Although the dissociation constants are identical, it is interesting to note that the association rate constants, derived from our BLI experiments, are different between the two antibodies. This could be correlated with the difference in the glycosylation pattern and the steric hindrance effect of slightly larger glycans ».
- We do not understand why no data points on the graph of Figure 6 were shown. The pdf version I have generated clearly shows the data points and the others referees did not make any comment regarding this issue.
- We have expanded the discussion by adding the FDA approved treatment against Ebola which consists of a cocktail of monoclonal antibodies produced in Nicotiana Benthamiana. Other examples of pharmaceutical treatments are cited in the previous paragraph.
- It is assume that the mammalian pattern is the natural one, and there is no effect here but should discuss that this may not always be the case
- I am not sure I understood this comment. There is no reason why it should be different for other antibodies as long as the same line of mutant seeds (FucT/XylT) is used for germination. We have an article which is currently under peer review showing the same homogeneity in the glycosylation partern with other kind of antibodies produced with Nicotiana Benthamiana plants obtained with DXF seeds.
We have added a sentence commenting this point in the discussion.
Reviewer 2 Report
Comments and Suggestions for Authors
The manuscript successfully demonstrates the feasibility and potential benefits of using plant-based systems for producing monoclonal antibodies in veterinary medicine. This study could serve as a pivotal reference for future research in the field and potentially pave the way for broader applications of plant-made pharmaceuticals in other areas of medicine.
The main focus of the report is looking into ways to use plants, specifically Nicotiana benthamiana, to make a recombinant form of canine IL-31 and an anti-IL-31 monoclonal antibody (M1). This approach is suggested as a potentially more cost-effective and rapid alternative to mammalian cell-based production. The report outlines the methodology of transient expression in plants effectively, making a compelling case for its advantages.
Introduction "Active and passive immunotherapies constitute a cornerstone for human therapies 37 of cancers and chronic diseases but have not been fully developed in veterinary medicine 38 yet. Veterinarians, scientists and dog owners hope that the strong interest in immuno- 39 therapy will benefit in improving canine health. Allergen-specific immunotherapy (de- 40 sensitization) s used in veterinary medicine since many decades to treat environmental 41 allergen-induded atopic dermatitis (reference required). Author should update with lastest references.
Despite the remarkable results of this therapeutic approach, there are 54 very few examples of passive immunotherapy in veterinary medicine. The scarcity of vet- 55 erinary Mab mainly comes from the general production cost of mAbs in mammalian cells and 56 therefore highlights the need of new platforms to produce them at a lower cost. For in- 57 stance, the current treatment of a 10 kg dog with a monthly injection of Lokivetmab costs 58 at least $ 1200 per year (references).
Materials methods : Okay
The seedlings were grown in a hydroponic system under continuous 111 LED lighting. why not use intermittent lighting?
Results and discussion: Okay
.
Author Response
We acknowledge the positive comments of this reviewer. We have responded to his/her comments in this revised manuscript.
- We have added the recent references in the introduction for allergen specific immunotherapy and for the cost of lokivetmab treatment.
The seedlings were grown in a hydroponic system under continuous 111 LED lighting. why not use intermittent lighting?
- Over the last 15 years we have checked several light regimes and the one presented here is the best to optimise plant biomass yield. Besides, we had information from Bergerac Seed and Breeding Company (France) that they undertake germination tests at 28°C with permanent light.
Reviewer 3 Report
Comments and Suggestions for Authors
Introduction: It is quite well-written and comprehensive.
- Line 40-41: Check the grammatical sentence
Materials and Methods: It is a well-described and easy-to-understand process, but it needs minor modifications;
- The animal health regulations must be well-stated with the reference or law's number.
- Line 232-233: work realized in duplicate or triplicate? Why it was not uniformized for the whole experiment?
- Line 239: For the In vivo experiment, the authors recorded the physical and biochemical parameters during the acclimatization, what was the interest for it if it didn't bring supplementary information about the animal's health conditions? and why not continue to examine during the proper experiment?
Results: good
Comments on the Quality of English LanguageThe article is quite novel revealing the possibility of replacing the large mammalian cell companies, involved in the storage and discovery of new treatments, with plant companies that are less expensive with easy control and short-time maintenance. Thus, the article could be considered for further publication if the authors improved some minor revisions.
Author Response
We acknowledge the positive comments of this reviewer and his/her recommendation for the publication of our manuscript. In our revised version, the different points raised by this referee are edited.
Line 40-41: Check the grammatical sentence
Correction was made
Materials and Methods: It is a well-described and easy-to-understand process, but it needs minor modifications;
The animal health regulations must be well-stated with the reference or law's number.
We have added the reference and all the details for the animal health regulations in Material and Methods.
This study was coded CE22-05 by the BIO 2M (Mezilles, France) ethics committee during its favorable evaluation authorized by the French Ministry of Research (26/12/2022) with the APAFIS code #39507-202211211340769 v5.
Line 232-233: work realized in duplicate or triplicate? Why it was not uniformized for the whole experiment?
These experiments were done with a Promega kit using well plate format. Therefore the number of samples to be analyzed was restricted. For this reason, we could not uniformized for the whole experiment.
Line 239: For the In vivo experiment, the authors recorded the physical and biochemical parameters during the acclimatization, what was the interest for it if it didn't bring supplementary information about the animal's health conditions? and why not continue to examine during the proper experiment?
We have added information in the material and methods about the physical and biochemical examination of the dogs during the study.